# Multimodal neuroimaging markers of variation in cognitive ability in older HIV+ men

**Ana Lucia Fernandez Cruz[1], Chien-Ming Chen[1], Ryan Sanford[1], D. Louis Collins[1], Marie-Josée Brouillette[2], Nancy E. Mayo[3], Lesley K. Fellows[1]***

**1** Department of Neurology and Neurosurgery, Montreal Neurological Institute, Faculty of Medicine, McGill University, Montreal, Quebec, Canada, **2** Department of Psychiatry, Faculty of Medicine, McGill University, Montreal, Quebec, Canada, **3** School of Physical and Occupational Therapy, Division of Clinical Epidemiology, Department of Medicine, McGill University, Montreal, Quebec, Canada

* lesley.fellows@mcgill.ca

## Abstract

### Objective

This study used converging methods to examine the neural substrates of cognitive ability in middle-aged and older men with well-controlled HIV infection.

### Methods

Seventy-six HIV+ men on antiretroviral treatment completed an auditory oddball task and an inhibitory control (Simon) task while time-locked high-density EEG was acquired; 66 had usable EEG data from one or both tasks; structural MRI was available for 43. We investigated relationships between task-evoked EEG responses, cognitive ability and immuno-compromise. We also explored the structural correlates of these EEG markers in the sub-sample with complete EEG and MRI data (N = 27).

### Results

EEG activity was associated with cognitive ability at later (P300) but not earlier stages of both tasks. Only the oddball task P300 was reliably associated with HIV severity (nadir CD4). Source localization confirmed that the tasks engaged partially distinct circuits. Thalamus volume correlated with oddball task P300 amplitude, while globus pallidus volume was related to the P300 in both tasks.

### Interpretation

This is the first study to use task-evoked EEG to identify neural correlates of individual differences in cognition in men living with well-controlled HIV infection, and to explore the structural basis of the EEG markers. We found that EEG responses evoked by the oddball task are more reliably related to cognitive performance than those evoked by the Simon task. We also provide preliminary evidence for a subcortical contribution to the effects of HIV infection severity on P300 amplitudes. These results suggest brain mechanisms and candidate biomarkers for individual differences in cognition in HIV.

**Data Availability Statement:** Data cannot be shared publicly because participants did not consent to this use of the data. The vulnerability of the population and the potentially sensitive nature

of the data were also considerations. Data are available for researchers who meet the criteria for access to confidential data; contact the authors or the MUHC Research Ethics Board to arrange access: cae@muhc.mcgill.ca.

**Funding:** LKF, MJB, NEM, DLC received grant funding from the Canadian Institutes of Health Research (TCO-125272) and its HIV Clinical Trials Network (CTN-273; CTN-026) to carry out this work. The funders had no role in study design, data collection and analysis, decision to publish, or preparation of the manuscript.

**Competing interests:** The authors have declared that no competing interests exist.

# Introduction

Despite effective viral suppression, cognitive impairment is a frequent concern in persons living with HIV, particularly as they grow older. Prevalence estimates are as high as 30 to 50%. [1, 2], although these may be over-estimates due to selection bias [3]. Executive function, attention, and processing speed are commonly affected in those with cognitive impairment [4, 5]. While typically mild in people taking combined antiretroviral therapy (cART), impairment can nonetheless limit occupational function, quality of life, and medication adherence [6–9], as well as the costs of medical care [10]. The causes and underlying neural mechanisms of cognitive impairment remain unclear. Candidate mechanisms include direct effects of HIV infection on the brain, especially prior to cART, cerebrovascular injury due to metabolic complications of chronic HIV, amongst others. There is no consensus on optimal neuroimaging markers or intervention targets.

Recent MRI studies have reported smaller brain volumes for subcortical nuclei and cortical regions in people with HIV infection, compared to HIV- groups [11–13]. However, the patterns of structural differences and their relationships with cognitive impairment have been inconsistent across studies [14] (Chang & Shukla, 2018). Whether brain dysfunction in HIV and the resulting cognitive impairment are the manifestations of processes affecting the whole brain or due to specific dysfunction in vulnerable regions or circuits has yet to be firmly established.

Structural imaging alone is unlikely to provide the answer. In contrast to neurodegenerative disorders such as Alzheimer's disease, many of the candidate pathophysiological mechanisms in HIV may not cause frank neuronal loss. Cytokine release due to neuroinflammation, for example, might impair synaptic plasticity or interfere with neurotransmission, without yielding macroscopic brain atrophy [15]. While there have been studies of brain function in HIV using functional MRI, less has been done with the simpler and more accessible functional imaging method of EEG [16]. Paired with appropriate cognitive tasks, EEG can assess specific neural circuit function with excellent temporal resolution. EEG is non-invasive and relatively inexpensive, so could be feasible even in resource-poor settings. There is evidence, albeit mainly from small samples, that measures from EEG (or the related technique of magnetoencephalography (MEG)) can distinguish HIV+ from HIV- groups, as well as groups with and without HIV-associated neurocognitive disorders (HAND) [16–19]. However, little work has been done to identify EEG markers associated with individual differences in cognitive performance amongst people with HIV; such markers could shed light on the underlying pathophysiology of cognitive impairment and serve as candidates for diagnostic or intervention-monitoring purposes at the individual level.

Here, we recorded stimulus-locked high-density EEG during performance of two cognitive tests: the Simon task to probe inhibitory control, and a demanding auditory oddball task requiring attention, vigilance and working memory, in the same sample of older men with well-controlled HIV infection. Structural MRI was also available in a sub-sample. In both tasks, we distinguished earlier EEG activity reflecting initial cortical sensory processing and conflict detection and later activity reflecting higher-order cognitive processes [20].

We aimed to provide evidence of the neural mechanisms underlying variation in cognitive performance in the normal to mild cognitive impairment range in men with HIV, seeking evidence for specificity at the level of neural circuits and earlier vs. later stages of cognitive processing. Demographic, metabolic, and HIV infection severity effects on the EEG markers were tested. In a sub-sample, structural MRI was used to localize cortical sources of EEG activity and to explore whether EEG results were explained by structural variation in candidate brain regions.

## Materials and methods

### Participants

Eighty-nine people living with HIV drawn from the Positive Brain Health Now (BHN) cohort participated in this study. BHN is a longitudinal study of brain health in people over age 35 living with HIV in Canada. Exclusion criteria were dementia precluding informed consent (Memorial Sloan Kettering severity $\geq$ 3 [21]), current substance dependence or abuse, psychotic disorder or non-HIV related neurological disorder, or terminal illness. Demographic and clinical data were acquired by self-report questionnaires, chart review and a brief cognitive test battery [22, 23]. We acquired high-density EEG in participants in two intervention sub-studies (trials of cognitive training or physical exercise) that drew from the BHN cohort (Fig 1). EEG was acquired as part of the baseline assessment prior to randomization into these intervention trials. These sub-samples were randomly drawn from BHN participants in Montreal, based on scores on a cognitive assessment battery. Sixty-nine participants were drawn from those with cognitive scores below the median, 20 from those above the median. Structural MRI was also available in 58 people in this sample. Although the BHN study included women, too few women (n = 6) were recruited into the intervention sub-studies (and thus completed EEG or MRI) to allow meaningful analysis by sex. Based on other work in this cohort, we expect differences in the causes and manifestations of brain dysfunction in men and women in this Canadian sample [9]. We thus report only on men here, to avoid falsely generalizing to women. The protocol was approved by the ethics board of the McGill University Health Centre and all participants provided written informed consent.

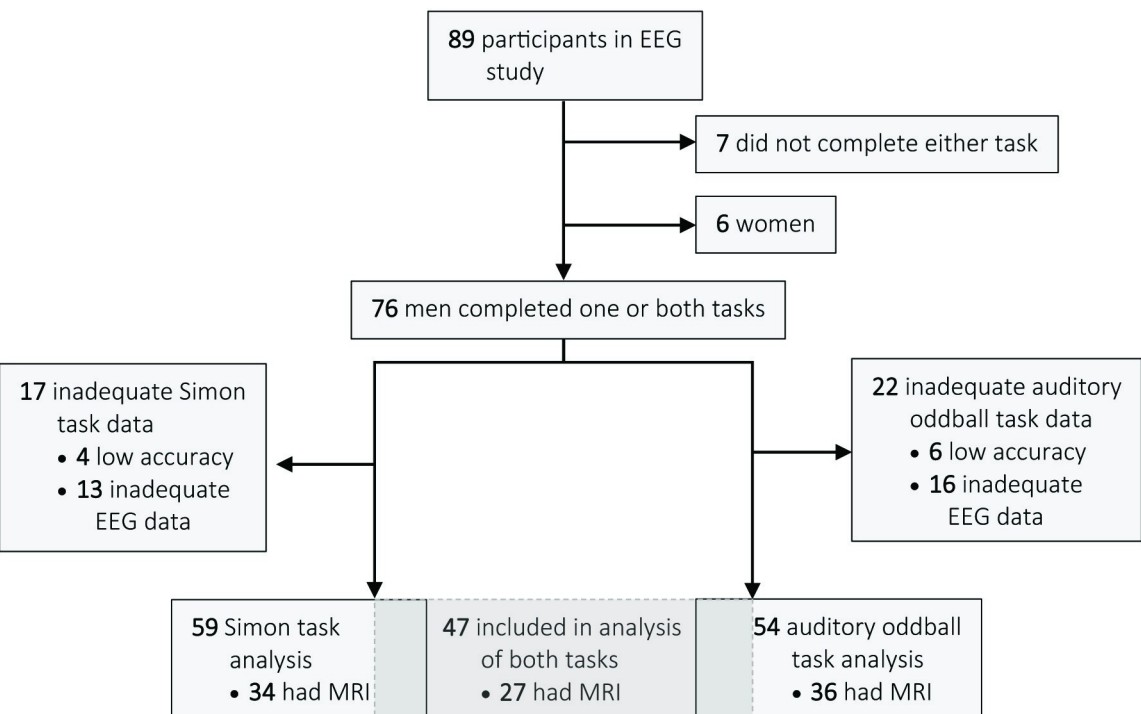

**Fig 1. CONSORT diagram of the study.** Participants who performed the task at chance accuracy or who had fewer than 25 usable EEG trials after pre-processing were excluded. Data were only available for 6 women, insufficient to reliably adjust for sex effects, so the analysis was restricted to men. Sixty-six men had complete data for at least one task, 47/66 had complete data for both tasks. Structural MRI was available in 43/66. Cognitive performance data were available for all participants.

## Cognitive and electrophysiological assessments

**Auditory oddball task.** The multiple auditory oddball task is shown in Fig 2. On each of 260 trials, participants fixated a cross on the computer screen and heard a series of 4 tones, the third of which might differ in frequency slightly (small deviant) or substantially (large deviant) from the other three. Participants reported whether this tone was the same or different with a left- or right-hand button press. Participants had a practice block with 28 trials before EEG recording began. The sensitivity index *d'* was calculated for the large and small deviant tones with the formula *d' = hit rate−false alarm rate* to estimate discrimination accuracy. Audiometry was carried out to ensure hearing adequate to perform the task.

**Simon task.** The Simon task required participants to make a left or right button press response according to an arbitrary sensorimotor rule: one of two shape cues was associated with the left button, the other with the right button (randomized across participants). The shape appeared on the left or right side of the screen, inducing a pre-potent response tendency to respond with that hand. The cue and side of presentation matched on congruent trials (190) and conflicted on incongruent trials (190) (Fig 3). Sixteen practice trials preceded the main task. Reaction time (RT) and accuracy were recorded.

**Cognitive assessment.** Cognitive ability was assessed with the Brief Cognitive Ability Measure (B-CAM), a computerized battery developed to assess domains typically affected in people with HIV, including processing speed, attention and cognitive control (Flanker task, Corsi block task), episodic memory (verbal list learning), and phonemic fluency (FAS). We

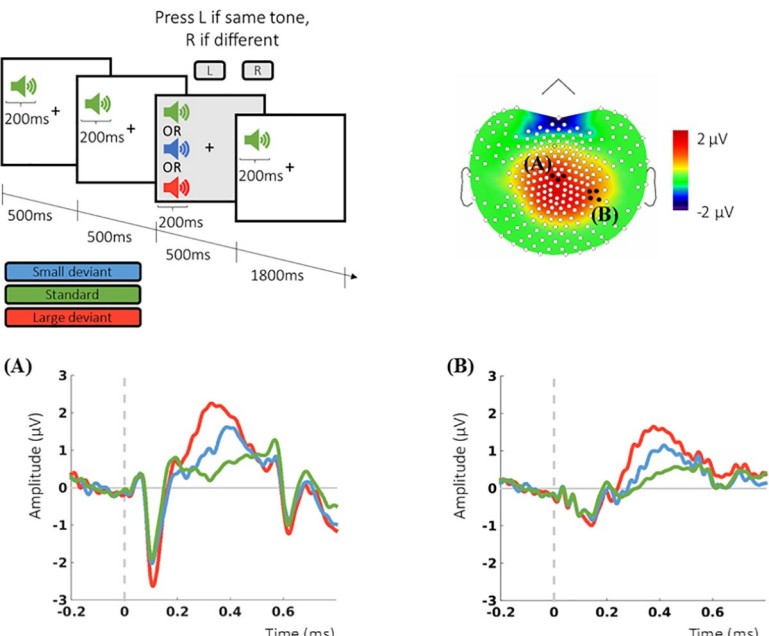

**Fig 2. Oddball task and evoked potentials.** Upper panel shows an example trial of the multiple auditory oddball task and the grand average topographic maps for the P300 with the 2 clusters of electrodes used for analysis highlighted (A and B). Participants completed 260 trials, each consisting of a sequence of four 200-milliseconds long tones presented binaurally at 70dB. The first, second and last tone of each of the trials were always "standard" tones (1047 Hz), the third tone could be either a standard tone or one of two deviant tones; a "small deviant" tone (1078 Hz) or a "large deviant" tone (1175 Hz). Participants were instructed to fixate a cross in the middle of the computer screen, and to report whether the third tone of each of the series was the same (right button press) or different (left button press) from the other three tones. Of the 260 trials, 146 (about 56% of all trials) were standard trials, 57 were small-deviant trials and 57 were large-deviant trials (each about 22% of all trials), presented in random order. (A) Grand average over the electrode cluster surrounding Cz and (B) over a cluster chosen following a data-driven approach.

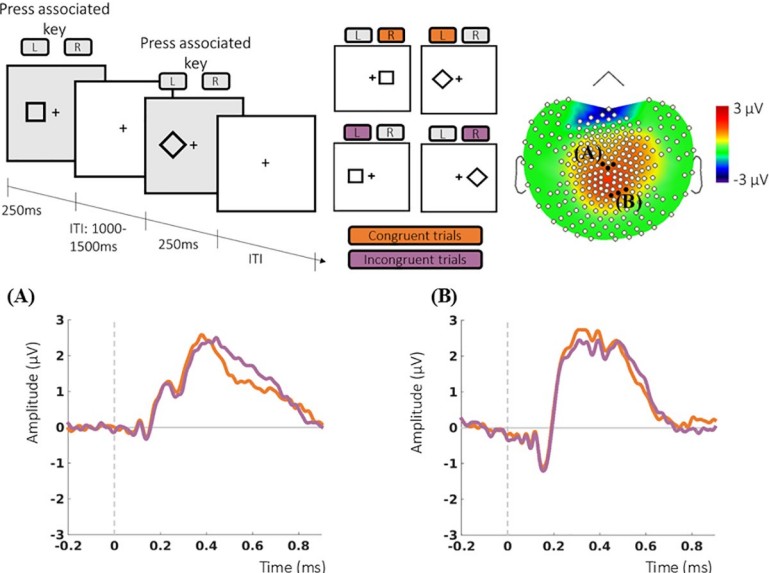

**Fig 3. Simon task and evoked potentials.** Upper panel shows example trials of the Simon task and the grand average topographic maps for the P300 with the two clusters of electrodes used for analysis highlighted (A and B). During the Simon task (upper right panel) participants responded to one of two 2.5 x 2.5 cm geometrical shapes (square or diamond) mapped to either a left or a right button press (counterbalanced across participants). Participants were instructed to press the button associated with the stimulus, regardless of where it appeared. There were 380 trials, 190 congruent (side of presentation and required response the same) and 190 incongruent (required response opposite to the side of presentation), presented in random order. Following the motor response, there was an inter-trial interval lasting 1 to 1.5 seconds (ITI). (A) Grand average over the electrode cluster surrounding Cz and (B) over a cluster chosen following a data-driven approach.

have shown that cognition can be assessed as a unidimensional construct in HIV [22, 24]. The B-CAM was developed using item-response theory as a measure of this latent construct. This battery has been validated against real world function (work status) and its concordance with multiple existing methods of operationalizing the Frascati classification of HAND has been reported [8]. Fifty-three participants included here also took part in another BHN sub-study that administered standard neuropsychological testing, covering six domains, at least two tests per domain [25], allowing HAND classification.

## Co-morbidity characterization

Self-reported depression and anxiety symptoms were assessed with the Hospital Anxiety and Depression Scale [26]. Risk of metabolic syndrome was calculated as a binary variable (risk or no risk). In accordance with the National Cholesterol Education Program (NCEP) Adult Treatment Panel (ATP) III guidelines, risk was defined if three or more diagnostic features of metabolic syndrome were present [27]. Features considered were: abdominal obesity (waist circumference $\geq$102 cm), elevated triglyceride levels ($\geq$1.7 mmol/L) or on lipid-lowering treatment, reduced HDL-C level ($<$1.0 mmol/L) or on lipid-lowering treatment, elevated blood pressure ($>$130/85 mmHg) or on antihypertensive treatment, and elevated fasting glucose ($\geq$5.6 mmol/L) or on medication for diabetes.

## Electroencephalographic recordings and analysis

EEG was recorded with a 256-channel HydroCel Geodesic-Sensor-Net (Electrical Geodesics, Inc., Eugene, OR) using Net Station 5 software. Data were collected using the electrode Cz as

reference and keeping electrode impedance below 50 kΩ. Seventy-eight electrodes located over the neck and cheeks were excluded before data pre-processing, as these were consistently contaminated with muscle artifacts.

EEG data were analyzed using Brainstorm [28]. Pre-processing followed standard procedures as recommended in [29] and Brainstorm. Continuous EEG recordings were filtered (0.1–30 Hz) down-sampled to 500 Hz and re-referenced to the right and left mastoid electrodes. Automatic blink detection was done using four electrodes located above and below each eye and artifact correction was performed with Signal-Space Projection.

Only correct trials without activity exceeding ± 100 μV were analyzed. The average number of trials included for the Simon task was 109 (SD 34) (congruent), and 102 (SD 38) (incongruent). For the oddball task, a mean of 51 (SD 8) large deviant and 49 (SD 9) small deviant trials were included.

**ERP analysis.**   Time windows were defined according to the maxima and time distributions of grand average waveform of the conditions of interest for each task. ERPs of interest for the oddball task were the N100 (mean amplitude between 90 and 120ms) and the P300 (mean amplitude between 280–400ms). For the Simon task, the analysis focused on the N200 (mean amplitude between 200–300 ms) and the P300 (mean amplitude between 300–550 ms). All epochs had a 200 ms pre-stimulus baseline.

Data extraction was twofold. First, clusters of electrodes centered on electrodes at which ERPs have a maximum distribution were used. The amplitude of the P300 was extracted at a cluster of electrodes (E081, E045, E132) centered on Cz [30, 31]. The amplitude of the N100 evoked by the oddball task was extracted from the cluster centered at Cz, and the N200 evoked by the Simon task at a cluster centered on frontocentral electrodes (E015 and E023) [32]. Additional extraction was in line with previous studies showing decreased P300 amplitude in cognitively impaired participants [33, 34]. This focused on the parieto-occipital and central-parietal clusters with maximum differences between conditions (corrected with False Discovery Rate at the 0.01 level) (Figs 2 and 3).

## MRI acquisition and analysis

MRI was acquired using a 3T Siemens Tim TRIO v17 scanner (Siemens AG, Erlangen, Germany) with a 12-channel transmit/receive head coil. The scanning protocol included a T1-weighted three-dimensional magnetization-prepared rapid acquisition gradient echo sequence (repetition time (TR)/echo time (TE)/inversion time (TI) = 2300/2.98/900ms; voxel = 1.0mm$^3$; flip angle = 9˚). The T1-weighted data were processed as described previously [35]. The pre-processing included denoising [36], intensity inhomogeneity removal [37] and brain extraction [38]. Linear registration of images was initially done in relation to the Montreal Neurological Institute ICBM152 template using a nine-parameter affine transform [39]. Linear registration was followed by a non-linear registration [40]. After registration, brain regions identified on the ICBM152 template [41] were mapped back to the subject's data to accurately identify specific structures of interest. The volumes in each of these regions were defined as the volume (cm$^3$) of all segmented voxels in the standard space. See also [12] for more details.

**Head modeling and ERP source estimation.**   ERP source estimation was carried out for the 43 participants who also underwent MRI. FreeSurfer (http://surfer.nmr.mgh.harvard.edu/) was used to extract and register cortical surfaces and white matter envelopes. Head modeling was performed using a symmetric boundary element method as implemented in the Open-MEEG software [42]. Minimum-norm imaging was used to estimate cortical sources of ERPs [43]. Current density maps and dipole orientations constrained to be normal to cortex were

used for inverse modeling. Z-score transformations were done to convert current density values at the subject level to a score representing the number of standard deviations with respect to the baseline period. Sources were rectified to absolute values and projected to default anatomy template before averaging across participants. Gaussian smoothing with a 3 mm full width at half maximum was applied to the final averages.

*Statistical analyses.* Statistical analyses were performed using R version 3.4.2. Repeated measures ANOVA with trial type as within subject factor was used to test for the effects of trial type on the task performance and ERPs of interest (N100, N200 and P300). This was done independently for each task using the factors small vs. large deviant vs. standard for the oddball task and congruent vs. incongruent for the Simon task. For the P300 effects–evaluated at two electrodes–electrode was included as an additional factor. To address our main question on the neural correlates of individual variation in cognitive ability in HIV+ individuals, we performed multiple linear regressions assessing the contribution of ERPs amplitude in predicting overall cognitive ability (B-CAM score).

Similarly, the effects of current and nadir CD4 cell counts on electrophysiological variables were assessed using multiple linear regression. Given that most participants had current CD4 cell counts in the normal range, and in line with recent MRI literature in cART-treated patients, it was hypothesized that the nadir CD4 count would be more relevant in predicting HIV-related EEG effects. Age and education (dichotomized as less than university, or at least some university education) were included in all models. Including educational effects at a finer grain did not contribute further, given the sample size and relatively high level of education of participants.

Pearson correlations were used to test whether volumes of subcortical nuclei previously shown to be affected in HIV and known to be involved in the cognitive tasks studied here (thalamus, globus pallidus and putamen) explained the variance observed in P300 amplitudes. Correlations between P300 amplitude and extra-cerebral cerebrospinal fluid (CSF) and total grey matter volumes were also examined to determine if diminished amplitudes were explained by overall brain volume or non-specific differences in conductance due to expanded CSF space between the cortex and scalp.

## Results

### Participant characteristics

Demographic and clinical variables are summarized in Table 1. All participants were taking cART and had intact hearing in the frequency range of the stimuli used in the oddball task. In the subset who underwent full neuropsychological testing, those meeting HAND criteria (asymptomatic neuropsychological impairment or mild neurocognitive disorder) had significantly lower BCAM scores (mean = 19.7, SD = 4.2) than those who did not meet HAND criteria (mean = 24.8, SD = 3.5), (t (51) = -3.2, p = 0.002).

### Behavioral performance

In the oddball task, large deviants were discriminated faster and more accurately than small deviants. Repeated measures ANOVA with trial type as within-subject factor showed a significant effect of trial type on RT (F (2, 52) = 60.79, p < 0.001, ε = 0.70, and d' (F (1, 53) = 14.06, p < 0.001, ε = 0.21). Post hoc Bonferroni-corrected pairwise comparisons showed that RTs were faster for large vs. small deviant (t (53) = -7.52, p < 0.001), and vs. standard trials (t (53) = -10.78, p < 0.001), as well as for standard vs. small deviants (t (53) = 4.51, p < 0.001). For the Simon Task, trial type had significant effects on reaction time (F (1, 57) = 80.56, p < 0.001, ε =

**Table 1. Demographic, clinical, and neuropsychological characteristics of the participants(N = 66).**

| Characteristic | |
|---|---|
| Age, mean (SD), y | 54 (7) |
| Education, No. (%) | |
| College diploma or less | 39 (59%) |
| Any university education | 27 (41%) |
| Cognitive Assessment | |
| B-CAM score, mean (SD) | 20.7 (4.2) |
| Neuropsychological HAND diagnosis | |
| Impaired (ANI or MND), No. (%) | 30 (56) |
| Anxiety (HADS) | 7.0 (4.5) |
| Depression (HADS) | 4.8 (4.0) |
| Presence of metabolic syndrome, No. (%) | 29 (44%) |
| Current CD4 cell count, median (IQR), cells/μL | 607 (477–852) |
| Nadir CD4 cell count, median (IQR), cells/μL | 170 (109–241) |
| Plasma Viral Load | |
| Virologically suppressed (<50 copies/mL), No. (%) | 63 (95%) |
| Duration of HIV infection, mean (SD), y | 18 (7) |

BCAM: Brief Cognitive Ability Measure; ANI: Asymptomatic Neuropsychological Impairment; MND: mild neurocognitive disorder; HADS Hospital Anxiety and Depression Scale; IQR: inter-quartile range; SD: standard deviation. The B-CAM is scored from 0 (worst cognitive performance) to 41 (best possible performance). B-CAM scores in this study ranged from 12 to 31.

0.58) and accuracy ($F_{(1, 57)} = 39.99$, $p < 0.00$, $\varepsilon = 0.41$), with participants less accurate and slower on incongruent trials.

## Task-evoked potentials

In the oddball task, the factors trial type ($F_{(2, 53)} = 42.51$, $p < 0.0001$, $\varepsilon = 0.45$), cluster of electrodes ($F_{(1, 53)} = 20.48$, $p < 0.0001$, $\varepsilon = 0.28$), and their interaction ($F_{(2, 53)} = 11.11$, $p < 0.0001$, $\varepsilon = 0.17$), had significant effects on the amplitude of the P300. Post-hoc pairwise comparisons (Bonferroni-corrected) showed that the amplitude of the P300 for the large deviant was significantly larger from that of the small deviant and the standard tone ($p < 0.0001$). The effect of trial type was also significant on the N100 ($F_{(2, 53)} = 18.87$, $p < 0.0001$, $\varepsilon = 0.26$), with the N100 evoked by the large deviant significantly different from the small deviant and standard tone ($p < 0.0001$). Evoked responses for the oddball task are shown in Fig 2.

In the Simon task, neither trial type nor cluster of electrodes had significant effects on the P300 or N200 amplitude. A significant interaction was found between these two factors ($F_{(1, 58)} = 9.40$, $p = 0.003$, $\varepsilon = 0.14$), but post-hoc pairwise comparisons showed no significant differences between the P300 amplitude of congruent and incongruent trials. Evoked responses for the Simon task are shown in Fig 3.

## Relationship between task performance and cognitive ability

The validity of both tasks as indicators of overall cognitive ability was confirmed using multiple linear regression to predict overall cognitive ability (B-CAM scores). In the oddball task, *d'* for the large deviant and age explained variation in B-CAM scores ($F_{(3,50)} = 10.3$, $p < 0.0001$). Neither RTs nor *d'* for the small deviant predicted cognitive ability. Thus, only large deviant trials were considered for further analysis.

In the Simon task, a significant relationship was found between incongruent trial RT and cognitive ability (F (3,55) = 4.56, p = 0.006). Neither congruent RT nor accuracy predicted B-CAM scores. Hence, the incongruent condition was retained for subsequent analysis.

### Relationships between task performance and HIV severity

Multiple linear regression showed that neither CD4 cell count (nadir, current), age nor education predicted performance of either task (all $p > 0.1$).

### Relationships between ERPs and cognitive ability

Multiple linear regression models were fitted for each task to evaluate if neural activity reflecting specific cognitive processes related to overall cognitive ability. B-CAM score was not significantly predicted when the amplitudes of early ERPs (N100 and N200) were included in the models ($p > 0.1$). In contrast, later ERPs were associated with overall cognitive ability in both tasks. The amplitude of the P300 evoked by the oddball task at the cluster of electrodes around Pz was significantly related to B-CAM (F (3, 50) = 4.73, p = 0.005), with a marginal effect of P300 amplitude ($p = 0.1$) and age ($p = 0.04$). At the central parietal cluster, the relationship was also significantly related to B-CAM (F (3, 50) = 6.60, p = 0.0007) (Table 2). In the Simon task, the relationship was similar at the cluster around Cz (F (3, 55) = 5.32, p = 0.002) and replicated at the second parieto-occipital cluster of electrodes: B-CAM score was predicted by the amplitude of the P300 (p = 0.0007) and age (p = 0.008) (F (3, 55) = 8.02, p = 0.0001) (Table 2).

### Relationships between ERPs and immunocompromise

For each of the tasks, linear regressions were carried out to test if current or past HIV infection severity contributed to the variation in neural activity in early or late stages of cognitive processing. The amplitudes of early ERPs were not significantly explained by nadir or current CD4 count. Nadir CD4 counts ($p = 0.0006$) and age ($p = 0.0001$) independently contributed to predicting the P300 amplitude evoked by the oddball task at the temporal cluster of electrodes, with an increase in 100 cells/μL in the nadir CD4 count related to an increase in amplitude of about 1 μV, (F (3,50) = 11.81, p < 0.000001). A similar significant contribution was found at the cluster centered on Pz ($p_{age} = 0.007$, $p_{nadir\_cd4} = 0.05$). Adding current CD4 count accounted only for an additional 3% of the variance observed in the P300 amplitude at the temporal cluster of electrodes. Neither nadir nor current CD4 counts explained the amplitude of the P300 evoked by the Simon task at either cluster of interest ($p > 0.9$) (Table 3; Fig 4).

**Table 2. Multiple linear regressions predicting cognitive ability (BCAM score).**

| Outcome: B-CAM, 20.7 (4.2) Predictors | Parameter Estimate (β) | Standard Error | 95% CI (lower bound, upper bound) | R² |
|---|---|---|---|---|
| **Oddball task** | | | | |
| P300 amplitude, μV | 1.16** | 0.43 | (0.31, 2.02) | |
| Age, decades | -1.17 | 0.83 | (-2.85, 0.48) | 0.28*** |
| **Simon task** | | | | |
| P300 amplitude, μV | 0.88*** | 0.25 | (0.38, 1.36) | |
| Age, decades | -1.78** | 0.65 | (-3.07, -0.47) | 0.30*** |

To facilitate interpretation age is expressed in decades. All regressions were corrected for education.

*$p < 0.05$

**$p < 0.01$

***$p < 0.001$.

**Table 3. Multiple linear regressions predicting ERP amplitudes.**

| Outcome, mean (SD), µV<br>  Predictors | Parameter Estimate (β) | Standard Error | 95% CI (lower bound, upper bound) | R² |
|---|---|---|---|---|
| | | Oddball task | | |
| N100 amplitude, -2.5 (2.1) | | | | |
|   Nadir CD4, 100 cells/µL | -0.13 | 0.19 | (-0.54, 0.26) | |
|   Age, decades | 0.11 | 0.40 | (-0.69, 0.92) | -0.04 |
| P300 amplitude, 1.5 (1.4) | | | | |
|   Nadir CD4, 100 cells/µL | 1.23**** | 0.09 | (0.25, 2.96) | |
|   Age, decades | -0.90*** | 0.22 | (-1.34, -0.46) | 0.41*** |
| | | Simon task | | |
| N200 amplitude, -0.70 (1.7) | | | | |
|   Nadir CD4, 100 cells/µL | -0.03 | 1.4 | (-3.26, 2.68) | |
|   Age, decades | 0.76** | 0.28 | (0.21, 1.32) | 0.17* |
| P300 amplitude, 2.2 (1.9) | | | | |
|   Nadir CD4, 100 cells/µL | 0.11 | 0.18 | (-0.26, 0.48) | |
|   Age, decades | -0.03 | 0.35 | (-0.7, 0.68) | -0.03 |

To facilitate interpretation, age is expressed in decades, and nadir CD4 count in 100 cells/µL. All regressions were corrected for education (college or less, or any university).

* $p < 0.05$

** $p < 0.01$

*** $p < 0.0001$.

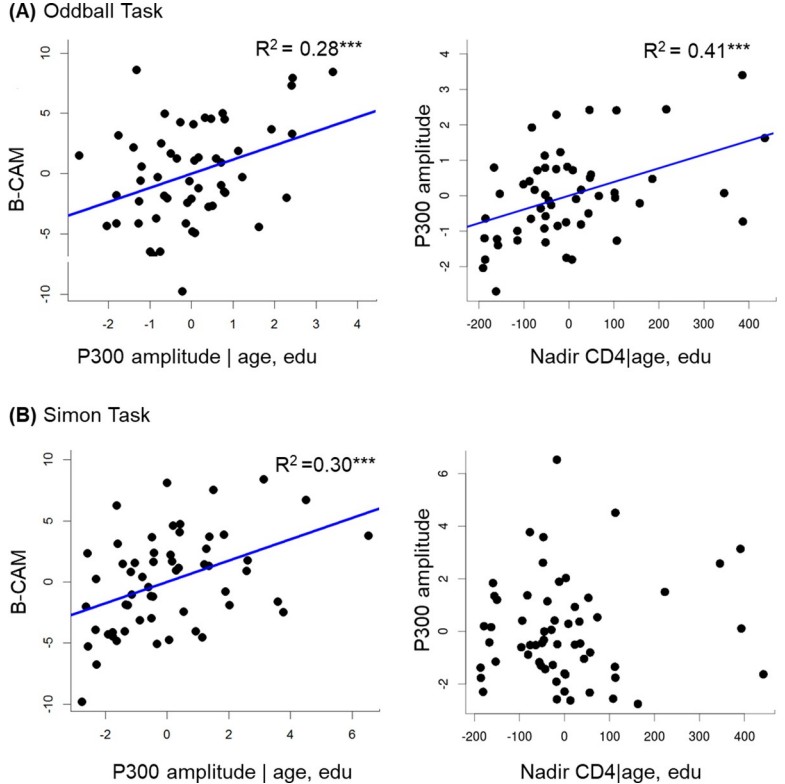

**Fig 4.** Partial regression plots of the relationship of P300 amplitudes evoked by (A) oddball and (B) Simon tasks with cognitive ability (B-CAM score) and immunocompromise (nadir CD4 cell count), controlling for the effects of age and education. *** $p < 0.001$.

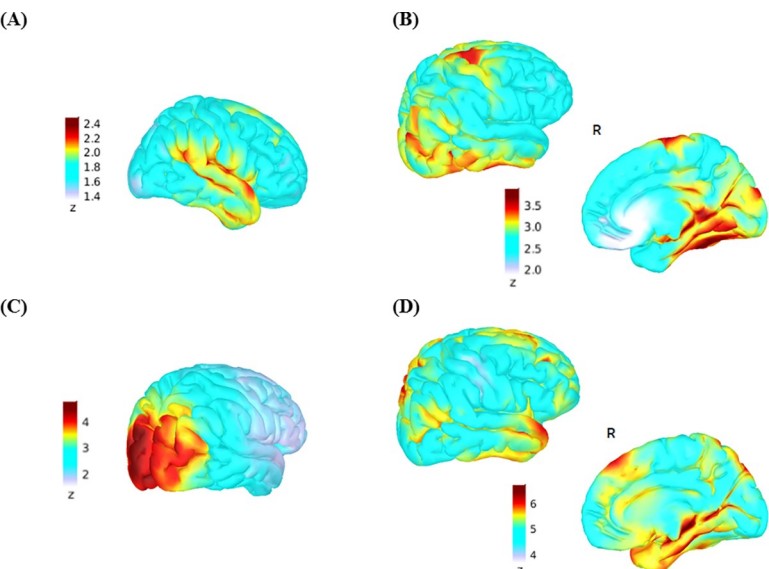

**Fig 5.** Source estimation for EEG activity in response to the oddball (A, B) and Simon tasks (C, D). Current density maps for both tasks were normalized in relation to values during the baseline period prior to stimulus presentation (-200 to 0 ms). Z-scores are with respect to the baseline period. (A) Current density maps during early auditory processing in response to the large deviant tone between 70 and 90 ms post stimulation. (B) Current density maps indicating sources for the P300 peak activations (280–400 ms) in response to the large deviant tone. The P300 evoked during the oddball task showed current density foci peaking in middle and inferior temporal gyri, posterior cingulate, temporal-parietal junction and the posterior superior frontal gyrus, principally on the right. (C) Early visual activity between 90–120 ms in response to incongruent stimuli. (D) Peak activations during the P300 time window in response to incongruent stimuli (300–550 ms). Lateralization to the right hemisphere also occurred for the P300 of the Simon task. As in the oddball task, activity localized to the temporal-parietal junction, but otherwise engaged different cortical regions: the anterior cingulate, middle and anterior-superior frontal gyrus and superior temporal gyrus. R = right hemisphere.

We repeated the linear regressions restricted to the participants who had completed both tasks (n = 47) to ensure that the observed differences in the relationship between nadir CD4 count and P300 amplitudes in the two tasks was not due to differences in the participants included in each analysis. The pattern of results remained unchanged; $p > 0.9$ for the Simon task and $p = 0.0001$ for the oddball task. We additionally tested if depression or anxiety, assessed with the HADS, or presence of metabolic syndrome explained any variance in the ERPs of interest. None of the regressions were statistically significant.

### Brain basis of evoked potentials

Fig 5 shows the EEG cortical source estimations for both tasks, in those for whom MRI data were available. Thalamus volume was related to the amplitude of the P300 evoked by the oddball task (oddball P300, $n = 27$, $r = 0.36$ $p = 0.03$) but not to the P300 evoked by the Simon task ($p's > 0.14$). The volume of the globus pallidus showed similar relationships with the P300 amplitude of both tasks (oddball P300, $n = 27$: $r = 0.32$, $p = 0.03$, Simon P300, $r = 0.46$, $p = 0.008$) (Fig 6). There was no correlation between P300 amplitude evoked by either task and putamen volume, nor were P300 amplitudes related to total grey matter volume or extra-cerebral CSF volume ($ps > 0.2$).

### Discussion

This study of the neural correlates of cognitive ability in middle-aged and older men with well-controlled HIV found evidence for specific vulnerabilities in processing stages and neural

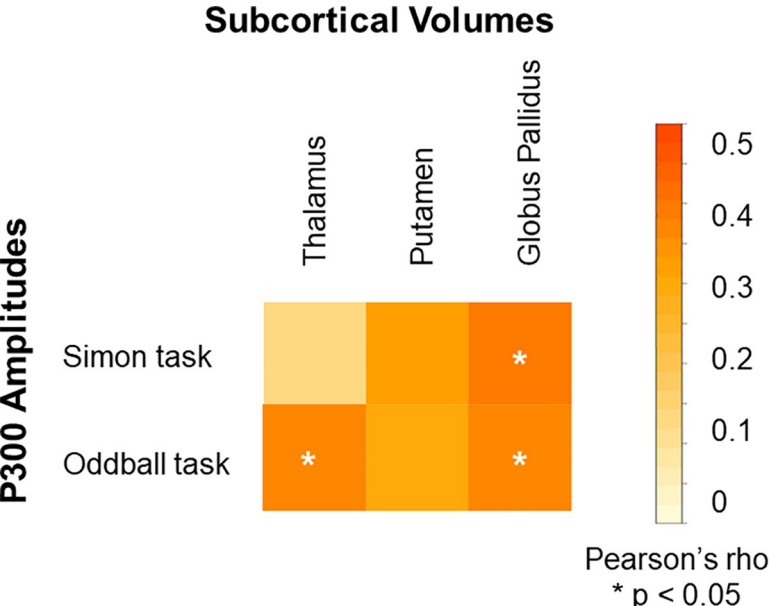

**Fig 6. Relationships between volumes of selected subcortical nuclei and P300 amplitudes in the Simon and oddball tasks.** Data are from the 27 participants with usable EEG data in both tasks and structural MRI. Asterisks indicate significant correlations (uncorrected). The scale indicates Pearson's r coefficients.

circuits. Behavioral measures from two tasks differing in higher-order response control and sensory-attentional requirements were related to overall cognitive ability (across a range from normal to mildly impaired) in this large, well-characterized sample. Stimulus-locked EEG activity evoked by these tasks at early sensory and cognitive processing stages was not significantly related to overall cognitive ability, nor was it associated with past HIV infection severity indicated by nadir CD4 count. In contrast, the amplitude of the P300 waveform evoked by both tasks did relate to cognitive ability assessed by a short battery of neuropsychological tests. In the auditory oddball task, P300 amplitude was also related to nadir CD4 count. Co-morbid depression, anxiety or metabolic syndrome did not explain variation in P300 amplitude in either task and ERP effects were not explained by global brain atrophy in the sub-sample who also underwent structural MRI. Instead, we found preliminary evidence implicating subcortical nuclei in P300 amplitude variation.

Prior smaller studies have compared early potentials evoked by auditory oddball and visual stimulation tasks in HIV+ and HIV- groups. Most reported no difference in N100 or N200 amplitudes between groups [44–46], although two studies found reduced N200 amplitudes in HIV+ groups [44, 47]. No work to date has asked whether early potentials relate to cognitive performance. Here, in a larger sample of HIV + individuals with varying degrees of cognitive ability, we show that these earlier processing stages (100–200 ms) in auditory and visual modalities do not reliably relate to cognitive performance nor to past HIV infection severity.

In contrast, cognitive ability was related to the amplitude of the P300 evoked by both tasks. Prior work on the P300 in HIV, mainly evoked with oddball tasks, found amplitude reductions in HIV+ compared to HIV- groups, in smaller samples with shorter duration of HIV infection than those studied here (reviewed in [16]). Here, taking an individual differences approach in men with HIV infection, we confirm the relevance of the P300 as an indicator of brain dysfunction in HIV, show for the first time that it relates to cognitive ability in this population, and find that this relationship is present across two different cognitive tasks in the same

sample. Smaller ERP amplitudes are a well-known feature of aging [48, 49], but the effects reported here were over and above those explained by age.

Neural activity in the P300 time window in the oddball task is thought to reflect attention towards working memory contents [31, 50] and arises primarily from the anterior cingulate, superior temporal gyrus, temporal-parietal junction and prefrontal cortex [51]. The Simon task P300 activity is thought to reflect action selection in the face of conflict, in addition to attention, as the task requires inhibition of prepotent responses [52]. This P300 originates principally from the middle and inferior frontal gyri and the superior parietal lobule [53]. Our source localization findings are largely in line with this literature in healthy subjects. We found that temporal-parietal junction, inferior temporal, pre-central and posterior regions of the superior frontal gyrus as well as posterior cingulate cortex contributed to P300 activity during the oddball task. P300 sources in the Simon task included middle and superior frontal gyrus, anterior cingulate, superior temporal gyrus and temporal-parietal junction. Thus, qualitatively, there were shared and distinct cortical sources for activity in this time window, across tasks, as intended.

The severity of past immunosuppression, indicated by the nadir CD4 count, explained variance in P300 amplitude in the oddball task only. There are likely multiple contributors to cognitive dysfunction in HIV [9]; this result suggests that P300 amplitude generally may be a good indicator of brain health, but the specific neural circuit underlying the auditory oddball task may be more vulnerable to the effects of HIV infection, at least in middle-aged and older men.

Taking advantage of the availability of MRI data for a subset of this sample, we provide preliminary evidence that this may be due to thalamic engagement in this task: smaller thalamus volumes were associated with smaller oddball P300 amplitude. The thalamus contributes to stimulus monitoring for subsequent evaluation according to context and working memory content and is engaged in auditory and visual oddball tasks [54, 55]. Recent neuroimaging studies using MRI, fMRI or PET have shown that the thalamus is amongst the structures most affected in HIV [12, 56–58], although this is not a universal finding [59]. Alternatively, this apparent circuit selectivity could reflect vulnerability of temporal-parietal cortex; the limited sample with MRI data available here does not allow a strong test of the latter possibility. In any case, these results suggest that weaker cognitive performance in HIV reflects more than the fronto-striatal dysfunction that has been a focus in the literature [60].

The relation between oddball P300 and nadir CD4 count, even after good immune recovery, may reflect brain injury sustained at the time of more severe HIV infection, or on-going effects of that injury, a so-called "legacy effect". Thus, early treatment initiation and on-going effective viral suppression may limit or prevent brain injury in persons living with HIV. Encouragingly, the few longitudinal neuroimaging studies to date show no strong evidence for progressive brain volume loss in cART treated individuals, at least over short periods [57, 61], in line with that idea.

A third explanation for the apparent circuit-specific vulnerability to HIV severity of the oddball task P300 that we observed here is that the waveforms evoked by these two tasks might differ in their measurement properties. While there were no ceiling or floor effects in either task, and the range and standard deviation of P300 amplitudes was similar, it could be that the signal-to-noise ratio of the Simon task-evoked P300 is lower than that of the oddball task, which would make it harder to detect a relationship with nadir CD4 count. Whether the reasons are technical or neurological, our findings argue that the auditory oddball task-evoked P300 may be more suitable for detecting HIV-related brain dysfunction than the P300 evoked by the Simon task, although both reflect cognitive ability.

Attenuation of P300 amplitude in auditory oddball and other tasks has been reported in other cognitively impaired populations, such as Alzheimer's disease and amnestic mild

cognitive impairment [33, 34], indicating that P300 amplitude reduction is not specific for HIV-associated neurocognitive disorders. EEG alone is thus better suited as a marker for brain dysfunction, rather than for identifying specific underlying causes. We also note that the P300 evoked in the present study is of smaller amplitude to those typical of the wider literature. This may be because the multiple oddball task here involves a series of tones, likely requiring greater attention to detect deviants than the conventional single tone task.

After accounting for age and education, nadir CD4 count only reliably explains some of the variance in P300 amplitude in the oddball task, and none in the Simon task. This emphasizes the need for more comprehensive identification of the factors contributing to variation in brain function in HIV. Dysfunction could be due to additional direct effects of HIV not indicated by nadir CD4 count, and many additional candidate mechanisms co-occurring or indirectly related to HIV status. Given sample size, we assessed a limited set of additional candidate mechanisms, finding no effect of anxiety, depression or metabolic risk (as a proxy for cerebrovascular injury) on the EEG measures. There may also be additional resilience factors beyond education [33]; for example, in other work, we have identified engagement at a high level in arts, music, sports or travel as indicators of cognitive reserve in older people with HIV [62].

This study has limitations. Foremost, the findings can be generalized only to middle-aged and older, relatively well-educated HIV+ men, on cART, and without clinically obvious dementia. There is a need to carry out similar work in women living with HIV. As only a subset of the sample had structural MRI, there was insufficient statistical power to carry out whole-brain analyses linking ERP and MRI, which was not an *a priori* aim of this study. Thus, the observed relationships between subcortical volumes and evoked potentials should be treated as exploratory and require replication in a new sample. Whole brain approaches would provide additional information on possible regional cortical or white matter contributors to the EEG variance, not tested here. Finally, we focused on variability in cognition amongst men with HIV; given this design, which did not include an HIV- group, we cannot claim that effects are HIV-specific, although the effects related to nadir CD4 count presumably reflect HIV-linked processes. In any case, our aim here was to identify EEG indicators of cognition in men with HIV, regardless of whether cognitive performance variability was uniquely determined by HIV status.

Other comorbidities common in HIV + individuals, such as cerebrovascular disease, cART neurotoxicity, lifestyle factors, or social exclusion may also contribute to brain dysfunction [63, 64]. Although the present study is amongst the largest EEG studies ever conducted in HIV, the sample size is nonetheless far from what would be needed to exhaustively test the effects of such factors.

## Conclusion

These are the first EEG results contrasting two tasks tapping different cognitive domains (with at least partially distinct underlying neural circuitry) in the same large, well-characterized sample of men with well controlled HIV infection. We showed that evoked potentials in the P300 time window are promising markers of cognitive ability in the mildly impaired-to-normal range of neuropsychological performance and suggest that tasks and EEG measures related to cortico-thalamic circuits may be particularly useful for probing brain health in people living with HIV. The immune effects of HIV at the time of untreated infection, indexed by nadir CD4 count, are related to brain function assessed by EEG even after many years of cART, and specific EEG measures in turn relate to current cognitive status. There appear to be additional, unmeasured contributors to EEG variance in these older men, beyond demographic variables;

neither anxiety, depression nor the presence of metabolic syndrome was explanatory here. It is increasingly clear that cognitive impairment in people living with HIV can have many contributors, at least some of which are likely to be treatable or remediable [9, 64, 65]. The findings here suggest that evoked potentials, a cheap and widely available readout of brain function, may prove useful in targeting or monitoring novel treatment or rehabilitation approaches to this common and quality-of-life-limiting aspect of living with HIV.

## Acknowledgments

We would like to thank Christine Déry and Susan Scott for their help with participant recruitment and data analysis, Marcus Sefranek for technical assistance and the Positive Brain Health Now investigators, research staff and participants for making this work possible.

## Author Contributions

**Conceptualization:** Ana Lucia Fernandez Cruz, Nancy E. Mayo, Lesley K. Fellows.

**Data curation:** Ana Lucia Fernandez Cruz, Ryan Sanford.

**Formal analysis:** Ana Lucia Fernandez Cruz, Ryan Sanford, D. Louis Collins, Nancy E. Mayo, Lesley K. Fellows.

**Funding acquisition:** Marie-Josée Brouillette, Nancy E. Mayo, Lesley K. Fellows.

**Investigation:** Ana Lucia Fernandez Cruz, Chien-Ming Chen, Nancy E. Mayo, Lesley K. Fellows.

**Methodology:** Ana Lucia Fernandez Cruz, Ryan Sanford, D. Louis Collins, Nancy E. Mayo, Lesley K. Fellows.

**Project administration:** Marie-Josée Brouillette, Nancy E. Mayo, Lesley K. Fellows.

**Resources:** Marie-Josée Brouillette, Nancy E. Mayo, Lesley K. Fellows.

**Software:** Ryan Sanford, D. Louis Collins.

**Supervision:** D. Louis Collins, Marie-Josée Brouillette, Nancy E. Mayo, Lesley K. Fellows.

**Validation:** Ana Lucia Fernandez Cruz.

**Visualization:** Ana Lucia Fernandez Cruz.

**Writing – original draft:** Ana Lucia Fernandez Cruz, Nancy E. Mayo, Lesley K. Fellows.

**Writing – review & editing:** Ana Lucia Fernandez Cruz, Marie-Josée Brouillette, Nancy E. Mayo, Lesley K. Fellows.

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
