## [Decision Letter · Decision Letter 0]

14 Jan 2021

PONE-D-20-36931

Multimodal imaging evidence for the brain basis of cognitive dysfunction in older HIV+ men

PLOS ONE

Dear Dr. Fellows,

Thank you for submitting your manuscript to PLOS ONE. After careful consideration, we feel that it has merit but does not fully meet PLOS ONE’s publication criteria as it currently stands. Therefore, we invite you to submit a revised version of the manuscript that addresses the points raised during the review process.

I agree with the reviewers that your study has two important limitations. On the one hand, there is no seronegative control group and, on the other hand, women were not included. This second limitation could be solved since you have data from 6 women, and it is not clear why they were excluded. The first requires discussion (you might consider including the explanation you give in your cover letter).

I invite you to make the necessary corrections based on the recommendations issued by the reviewers. In particular, I consider it necessary to clarify the modification of the tasks and increase the information about the participants. Years of schooling have been used as a proxy of the cognitive reserve; if this information was available, it could be explored whether a greater cognitive reserve has attenuated the effects of the disease on cognitive function (see Gu et al. among your references) and, perhaps, eliminate the effect of years of schooling in the model (using a quantitative variable instead of a dichotomous categorical variable).

We look forward to receiving your revised manuscript.

Kind regards,

Thalia Fernandez, Ph.D.

Academic Editor

PLOS ONE

Journal Requirements:

'Funding

This work was supported by grants from the Canadian Institutes of Health Research, TCO-125272, the CIHR Canadian HIV Trials Network, CTN 273 (LKF, M-JB, NEM), and support from the Fonds de Recherche Santé du Québec (LKF), the McGill Integrated Programme for Neuroscience Research Institute of the McGill University Health Centre (MJB) and the McGill Integrated Programme for Neuroscience (ALFC).'

'LKF, MJB, NEM, DLC received grant funding from the Canadian Institutes of Health Research (TCO-125272) and its HIV Clinical Trials Network (CTN-273; CTN-026) to carry out this work. The funders had no role in study design, data collection and analysis, decision to publish, or preparation of the manuscript.'

Reviewers' comments:

Reviewer's Responses to Questions

**Comments to the Author**

1. Is the manuscript technically sound, and do the data support the conclusions?

Reviewer #1: Partly

Reviewer #2: Yes

2. Has the statistical analysis been performed appropriately and rigorously? 

Reviewer #1: Yes

Reviewer #2: Yes

3. Have the authors made all data underlying the findings in their manuscript fully available?

Reviewer #1: No

Reviewer #2: Yes

4. Is the manuscript presented in an intelligible fashion and written in standard English?

Reviewer #1: Yes

Reviewer #2: Yes

5. Review Comments to the Author

Reviewer #1: This exploratory study examines 66 men with HIV on cART utilizing EEG and sMRI. The authors use two cognitive tasks during EEG acquisition: a simon task, and an auditory oddball task. They go on to examine time-domain signal properties and confirmatory perform source localization on their EEG data, and extract subcortical volumes from their sMRI data. They find that P300 is related to global cognition, and P300 from the oddball is related to CD4 nadir. Strengths include multimodality (although this only uses a small subsample of n=27) and examination of two EEG tasks. However, their clear limitation is a lack of an uninfected control group. This limits their HIV related conclusions to the relationship between P300 and CD4 nadir. Further issues include only examining men, limited research question examining small variation in cognitive function, and an analysis that appears somewhat selective.

Abstract

The abstract sample size is misleading, especially for the multimodal portion, which they state 54 completed MRI, but ultimately only used 27 of those participants.

“This is the first study to combine structural and functional imaging in an overlapping sample to address the neural circuits related to cognitive dysfunction in HIV.” Although relatively few studies exist, this isn’t the first. The following is just a selection of studies all have structure, function and cognitive testing in a single sample: (Cole, James H., et al. Clinical Infectious Diseases 66.12 (2018): 1899-1909; Samboju, Vishal, et al. NeuroImage: Clinical 20 (2018): 327-335; Wilson et al., Human brain mapping 36.3 (2015): 897-910.)

Introduction

While cognitive impairment is certainly still an issue in older men with HIV, I think the authors should acknowledge some of the recent literature calling into question whether the prevalence of HAND has been overestimated (Su et al., Aids 29.5 (2015): 547-557; Zheng et al., Aids 33.14 (2019): 2115-2124; Meyer et al., Neuroepidemiology 41.3-4 (2013): 208-216).

I’m not sure I would categorize an auditory oddball task as a “attention and working memory-requiring categorization task.” Although true, that is likely true of any task, and the inherent contrast of oddball tasks doesn’t probe working memory, moreso vigilance.

The authors state their goal was to “provide evidence of the neural mechanisms underlying variation in the mild cognitive impairment to normal range.” Further justification for this aim is needed as I’m left wondering why mild (asymptomatic) to normal fluctuations in cognitive function is relevant, as opposed to more severe cognitive impairment. Their lone justification is in their first paragraph stating “impairments can nonetheless limit everyday function.” I encourage them to expand upon this further, potentially adding more about the papers they cite relating cognitive function to medication adherence and quality of life.

Methods

It seems inappropriate to have excluded women from the study, especially when data were available for 6 women. While the authors state a sample of 6 would be insufficient to examine sex effects, the relevant research question here does not necessitate the exclusion of participants based solely on sex.

The entire analysis is based on a global composite of the B-CAM. They state 53 participants had standard neuropsychological testing. What to their results look like examining these participants and their standard neuropsychological testing data?

The authors cite away the entire structural analysis. A brief summary of the segmentation in this manuscript would be helpful.

Was a variable inter-stimulus interval used? This is important considering the prestimulus baseline.

Results

The sample had an average B-CAM of 20.7. Looking into the B-CAM, it appears the maximum score is a 24. There may be ceiling effects at play. Can the authors provide a plot showing the distribution of B-CAM scores?

It is stated that evoked potentials displayed in Figures 2 and 3 represent grand averages, which would imply an average across the sample. It would therefore be helpful to show error bars on these timeseries.

Much of the data is thrown out early on in the analysis. Only large deviant trials and incongruent trials are used for the HIV analysis based on the relationship between behavior on these trials and BCAM scores. I believe the more intuitive approach would be to utilize contrasts between conditions. The authors even show behavioral effects between conditions in their repeated measures ANOVAs. Why then are these condition effects suddenly ignored when examining neural data?

There appears to be a typo, with effect of P300 amplitude stated twice with differing p values: “The amplitude of the P300 evoked by the oddball task at the cluster of electrodes around Pz was significantly related to B-CAM (F (3, 50) = 4.73, p = 0.005), with a marginal effect of P300 amplitude (p = 0.1) and age (p = 0.04).”

Given the use of a global B-CAM score, can the authors provide any insight into whether certain cognitive domains drove these differences?

The source localization analysis seems more supplementary. It does not give added information besides that the P300 for the two tasks are not perfectly identical when source localized, which I wouldn’t expect in the first place given the large differences in the tasks. The maps actually looked more similar than I would have expected.

Why examine only the 27 participants that completed both tasks and MRI, when separate correlations are run for each task?

Given the relevant results are derived from regressions and correlations, scatterplots with best fit lines for all significant results are needed to get a better sense of the distributions of the variables.

Discussion

The authors state: “provided evidence that early sensory and cognitive processing stages in both tasks did not relate to overall cognitive ability…” Technically their statistics did not test for support of the null hypothesis, but just failed to reject the null hypothesis. Baysean statistics could be used to show that these variables in fact do not relate to cognitive ability ect.

Additionally they state: “Here, in a larger sample, using cortical source estimation we show that these earlier processing stages in auditory and visual modalities are relatively spared.” This is not accurate because they do not have an uninfected control group to compare to. They also do no use source estimation other than to qualitatively compare P300 of the two tasks.

Further acknowledgment of other health comorbidities needs to be addressed in the limitations, as even factors such as adipose tissue can be related to neural alterations that might have otherwise been attributed to HIV (Lake et al., 2017. J Neurovirol;23(3):385-393). I use this as an example, as a multitude of other comorbidities are not explicitly controlled for in this study, and may therefore be contributing factors.

Reviewer #2: This is an interesting yet straightforward investigation of the association of event related potentials, recorded during an auditory oddball task and visual Simon task, with nadir CD4 count and cognitive ability among older HIV-1 seropositive men. There is, unfortunately, no HIV-1 seronegative control group and no women were included. But, the findings are consistent with the findings of similar studies published previously. I think that this MS is clearly written. It includes appropriate analyses. The conclusions are reasonable and justified by the findings.

I do have a few minor comments and suggestions:

1. Both the auditory oddball and Simon tasks used presently are different in format from what we usually see. Here, the auditory task is not a running series. It presents the deviant stimulus at a predictable time (#3 in a series of 4) and therefore produces a smaller P300 than we typically see in this literature. This should be pointed out to readers not familiar with ERP methods. Also, the Simon task is not presented in a typical manner.

2. It would be useful to know more about the participants. What was the race/ethnic composition of the sample? Instead of telling us the percentage who attended college, please report the number of years of education and the variability? Do you have any information about family history, substance abuse history, or history of childhood ADHD or conduct problems? As the authors know, all of these factors have been shown to affect P300 and complicate the interpretation of HIV effects on P300.

3. It is not surprising that anxiety and depression were not related to ERPs in this sample. First, it is a prospectively followed cohort that may be healthier and therefore more cooperative than we see in cross-sectional studies. Second, no women were studied--rates of anxiety and depression are generally lower in men.

4. In addition to citing your recently published review of P300 (and other ERP components) in HIV, you might cite a few of the original primary sources.

6. PLOS authors have the option to publish the peer review history of their article (what does this mean?). If published, this will include your full peer review and any attached files.

Reviewer #1: No

Reviewer #2: No

---

## [Author Response · Author response to Decision Letter 0]

7 Apr 2021

The manuscript has been revised to address the specific points raised by reviewers, as described in the attached Response to Reviewers document.

In addition, we appreciate the Editor's thoughtful comments and respond as follows:

1. The lack of an HIV- group is not, in our opinion, a limitation of this study. We do not aim to isolate effects unique to people with HIV, but rather seek to identify EEG responses that relate to cognitive ability amongst people with HIV. In this revision, we clarify this goal, which we believe to be a relevant approach to identifying candidate biomarkers for cognition in this avowedly multifactorial clinical presentation. Having identified EEG markers, we go on to probe the factors that influence those markers, some HIV-specific (nadir CD4) and some generic: demographics, metabolic syndrome, anxiety and depression. We think that much of the literature on EEG in HIV is limited by very small samples and control groups that vary in many ways from the HIV+ group, beyond sero-status. Indeed, the study of cognition in HIV has been dogged by the challenges of choosing appropriate control groups. Our approach gets away from this problem, pursuing aims that we argue are more relevant to identifying biomarkers of cognitive impairment.

2. We now explain at more length why we have not included women in this analysis. We have extensive data from the parent cohort suggesting that women are likely to differ in the brain underpinnings of cognitive performance in our sample. Women recruited to our cohort are systematically different in many ways from men in our cohort, reflecting the demographics of HIV in Canada. We think it is better to be clear about the inability to generalize the findings here to women, than to include an inadequate sample of women and pretend otherwise. We agree that appropriately powered research on women is urgently needed and are currently recruiting for such a study. We make these points in the manuscript.

3. We also agree that more nuanced work on cognitive reserve in HIV is of value. We have studied cognitive reserve in HIV and appreciate the challenges of fully capturing this much-debated construct. Given the sample size available here, we think adding a more complex cognitive reserve indicator is unlikely to shed much more light. We preliminarily explored other indicators of cognitive reserve (and more granular consideration of education), and this did not add substantially to the simple consideration of education that we included a priori. We are concerned that this study is already presenting very rich data, with a novel approach to the overall design, and a novel cognitive assessment measure, as well as multimodal imaging and two behavioral tasks, and that introducing an unplanned, complex approach to cognitive reserve in addition will only confuse readers. We have added a comment on cognitive reserve as a future direction of interest in the Discussion.

---

## [Decision Letter · Decision Letter 1]

24 Jun 2021

PONE-D-20-36931R1

Multimodal neuroimaging markers of variation in cognitive ability in older HIV+ men

PLOS ONE

Dear Dr. Fellows,

Thank you for submitting your manuscript to PLOS ONE. After careful consideration, we feel that it has merit but does not fully meet PLOS ONE’s publication criteria as it currently stands. Therefore, we invite you to submit a revised version of the manuscript that addresses the points raised during the review process.

I think the article could be accepted after it satisfies the following requests from Reviewer 1:

1) Please indicate the final size of the multimodal subsample of 27 in the abstract, as the title emphasizes multimodality, and the results in the abstract refer specifically to the correlations of the n = 27 subsample analysis.

2) Include partial regression scatterplots in some way, potentially as supplemental material.

We look forward to receiving your revised manuscript.

Kind regards,

Thalia Fernandez, Ph.D.

Academic Editor

PLOS ONE

Journal Requirements:

Additional Editor Comments (if provided):

Dear Dr. Fellows,

I consider the article could be accepted after it satisfies the following requests by the Reviewer 1:

1) Stating the final multimodal subsample size of 27 in the abstract since the title emphasized multimodality, and the results in the abstract specifically reference correlations from the analysis of the n=27 subsample.

2) Including the partial regression scatter plots somehow, potentially as supplemental material.

Kind regards,

Reviewers' comments:

Reviewer's Responses to Questions

**Comments to the Author**

1. If the authors have adequately addressed your comments raised in a previous round of review and you feel that this manuscript is now acceptable for publication, you may indicate that here to bypass the “Comments to the Author” section, enter your conflict of interest statement in the “Confidential to Editor” section, and submit your "Accept" recommendation.

Reviewer #1: All comments have been addressed

Reviewer #2: All comments have been addressed

2. Is the manuscript technically sound, and do the data support the conclusions?

Reviewer #1: Yes

Reviewer #2: Partly

3. Has the statistical analysis been performed appropriately and rigorously? 

Reviewer #1: Yes

Reviewer #2: Yes

4. Have the authors made all data underlying the findings in their manuscript fully available?

Reviewer #1: No

Reviewer #2: Yes

5. Is the manuscript presented in an intelligible fashion and written in standard English?

Reviewer #1: Yes

Reviewer #2: Yes

6. Review Comments to the Author

Reviewer #1: The authors have responded to my extensive comments. Many of the responses and revisions I fully agree with and appreciated. While I still hold some minor disagreements, ultimately the paper is improved and further rebuttal is likely unnecessary.

My remaining two requests include:

1) Stating the final multimodal subsample size of 27 in the abstract since the title emphasized multimodality, and the results in the abstract specifically reference correlations from the analysis of the n=27 subsample.

2) Including the partial regression scatter plots somehow, potentially as supplemental material.

Should these be done, I feel the paper is acceptable for publication.

Reviewer #2: (No Response)

7. PLOS authors have the option to publish the peer review history of their article (what does this mean?). If published, this will include your full peer review and any attached files.

Reviewer #1: No

Reviewer #2: No

---

## [Author Response · Author response to Decision Letter 1]

5 Jul 2021

I think the article could be accepted after it satisfies the following requests from Reviewer 1:

Thanks for the opportunity to make final adjustments to this submission to address the remaining concerns of Reviewer 1.

1) Please indicate the final size of the multimodal subsample of 27 in the abstract, as the title emphasizes multimodality, and the results in the abstract refer specifically to the correlations of the n = 27 subsample analysis.

We have added this information to the abstract. Slight edits were therefore made to the abstract as a whole in order to keep within the word limit.

2) Include partial regression scatterplots in some way, potentially as supplemental material.

We have included these scatterplots as Figure 4. Remaining Figures were renumbered.

---

## [Editor Report · Decision Letter 2]

12 Jul 2021

Multimodal neuroimaging markers of variation in cognitive ability in older HIV+ men

PONE-D-20-36931R2

Dear Dr. Fellows,

We’re pleased to inform you that your manuscript has been judged scientifically suitable for publication and will be formally accepted for publication once it meets all outstanding technical requirements.

Kind regards,

Thalia Fernandez, Ph.D.

Academic Editor

PLOS ONE
---

## [Editor Report · Acceptance letter]

19 Jul 2021

PONE-D-20-36931R2 

Multimodal neuroimaging markers of variation in cognitive ability in older HIV+ men 

Dear Dr. Fellows:

I'm pleased to inform you that your manuscript has been deemed suitable for publication in PLOS ONE. Congratulations! Your manuscript is now with our production department. 

Kind regards, 

on behalf of

Dr. Thalia Fernandez 

Academic Editor

PLOS ONE